# Kinetic control of tunable multi-state switching in ferroelectric thin films

R. Xu[1], S. Liu [2], S. Saremi [1], R. Gao[1], J.J. Wang[3], Z. Hong[3], H. Lu [1], A. Ghosh [1], S. Pandya[1], E. Bonturim [1], Z.H. Chen[1,4], L.Q. Chen[3], A.M. Rappe [5] & L.W. Martin [1,6]

Deterministic creation of multiple ferroelectric states with intermediate values of polarization remains challenging due to the inherent bi-stability of ferroelectric switching. Here we show the ability to select any desired intermediate polarization value via control of the switching pathway in (111)-oriented $PbZr_{0.2}Ti_{0.8}O_3$ films. Such switching phenomena are driven by kinetic control of the volume fraction of two geometrically different domain structures which are generated by two distinct switching pathways: one direct, bipolar-like switching and another multi-step switching process with the formation of a thermodynamically-stable intermediate twinning structure. Such control of switching pathways is enabled by the competition between elastic and electrostatic energies which favors different types of fer-roelastic switching that can occur. Overall, our work demonstrates an alternative approach that transcends the inherent bi-stability of ferroelectrics to create non-volatile, deterministic, and repeatedly obtainable multi-state polarization without compromising other important properties, and holds promise for non-volatile multi-state functional applications.

[1] Department of Materials Science and Engineering, University of California, Berkeley, CA 94720, USA. [2] Extreme Materials Initiative, Geophysical Laboratory, Carnegie Institution for Science, Washington, DC 20015, USA. [3] Department of Materials Science and Engineering, Pennsylvania State University, State College, PA 16802, USA. [4] School of Materials Science and Engineering, Harbin Institute of Technology, 518055 Shenzhen, China. [5] Department of Chemistry, University of Pennsylvania, Philadelphia, PA 19104-6323, USA. [6] Materials Sciences Division, Lawrence Berkeley National Laboratory, Berkeley, CA 94720, USA. Correspondence and requests for materials should be addressed to L.W.M. (email: lwmartin@berkeley.edu)

Ferroelectric materials with electrically addressable polarization hold great promise for non-volatile and low-power device operation[1,2]. This promise has been partially realized based on reversible bipolar switching in ferroelectrics, which has enabled numerous devices including non-volatile memories[3], low-power logic[4,5], and nanoscale sensors and actuators[6,7]. To address the increasing demands of computational density and functionality for next-generation electronics, it has become clear that if one can overcome the inherent bi-stability of ferroelectrics to create deterministic multi-state polarization (i.e., many states with volume-averaged polarization values between up and down), a new generation of devices becomes possible, including increased memory storage density, adaptive-computational systems, and much more[8,9].

Due to the inherent bi-stability of ferroelectric switching, deterministic creation of multiple stable states with distinct polarization values (henceforth multi-state polarization) remains a hallmark challenge. Efforts in this regard, which rely on controlling the fraction of switched domains to access multi-state polarization, fail to precisely and repeatedly achieve the same fraction of switching. This arises from the stochastic nature of inhomogeneous nucleation in ferroelectric switching and is further exacerbated in small material volumes, preventing device downscaling. Recent studies have attempted to introduce external factors such as defects and non-switchable interfacial layers to gain more control over the nucleation process[10–12], but progress in this regard comes at the cost of degrading the ferroelectric properties including worsening fatigue[13–16] and imprint properties[17,18]. Prior work has also attempted to extrinsically achieve multi-state polarization by regulating the displacement current flow during polarization switching[19], which does not rely on particular material structures.

Other approaches to generate multi-state polarization have focused on the design and control of the architecture of the ferroelectric. For instance, prior work has demonstrated that it is possible to create multi-stability in ferroelectrics by controlling the number of switched ferroelectric layers in multi-layer heterostructures[20,21] or by controlling sequential-polarization switching in films with multiple coexisting structural instabilities[22–25]. These efforts, which rely on a single switching pathway between possible structural variants, usually lead to a limited number of switching states that are dictated by the number of film layers or available structural variants. Thus far, an alternative approach that transcends the inherent bi-stability of ferroelectrics to create non-volatile, deterministic, and repeatedly obtainable multi-state polarization without compromising other important properties, has not been achieved.

Here, instead of engineering the number of available structural variants, we provide a novel approach to achieve multi-stability in ferroelectrics by engineering the number of switching pathways between these structural variants. In particular, we demonstrate that it is possible to create (at least) 12 polarization states in a single (111)-oriented $PbZr_{0.2}Ti_{0.8}O_3$ layer via kinetic control of two distinct switching pathways. Combining thin-film growth, capacitor-based pulse-switching studies, molecular-dynamics (MD) simulations, phase-field modeling, and piezoresponse force microscopy (PFM), the mechanism underlying the realization of the multi-state polarization is revealed to be a previously unobserved process wherein the different polarization states are associated with variations in the volume fraction of two geometrically distinct domain twinning structures generated by the competition of two switching pathways: one direct, bipolar-like switching and another multi-step switching process with the formation of a thermodynamically-stable intermediate twinning structure. The resulting domain structures provide repeatable, deterministic, and stable multi-state polarization beyond that observed to date.

## Results

**Domain structure characterization.** This work focuses on single-phase and epitaxial (111)-oriented $PbZr_{0.2}Ti_{0.8}O_3$ heterostructures grown via pulsed-laser deposition (Methods)[26,27] with nearly ideal hysteresis behavior (Supplementary Figs. 1 and 2). PFM studies were performed on these heterostructures to understand the static domain structure. For a (111)-oriented tetragonal ferroelectric thin film, the polarization can point in any of six, energetically-degenerate directions (i.e., three pointing out of the plane of the film along the [100], [010], and [001], corresponding to $P_1^+$, $P_2^+$, and $P_3^+$, respectively, and three pointing into the plane of the film along the $[\bar{1}00]$, $[0\bar{1}0]$, and $[00\bar{1}]$, corresponding to $P_1^-$, $P_2^-$, and $P_3^-$, respectively; Fig. 1g). Upon applying a positive voltage (8 V) to fully pole the film using a PFM tip, we observe the formation of multiple unique, but related ordered domain structures which exhibit similar twinning configurations but varying orientations (Fig. 1a–c). Further analyses reveal that each of these three domain structures possess $P_1^-$, $P_2^-$, and $P_3^-$ variants (combined in a 1:1:2, 1:2:1, or 2:1:1 ratio for Fig. 1a–c, respectively) which are tiled into three different twinning configurations and result in a fully (down) poled net polarization. Similarly, applying a negative voltage ($-8$ V) to the heterostructures, we also observe the same type of twinning configurations but with the opposite out-of-plane polarized variants ($P_1^+$, $P_2^+$, and $P_3^+$). We term these domain structures as Type-I twinning structures. Applying an intermediate voltage ($-4$ V) to a fully down-poled Type-I twinning structure, however, results in the formation of another type of ordered superdomain structure which possesses $P_1^-$, $P_2^+$, and $P_3^-$ in a 1:2:1 ratio (Fig. 1d), $P_1^-$, $P_2^-$, and $P_3^+$ in a 1:1:2 ratio (Fig. 1e), or $P_1^+$, $P_2^-$, and $P_3^-$ in a 2:1:1 ratio (Fig. 1f), all possessing zero net out-of-plane polarization. Similarly, applying 4 V to the up-poled Type-I twinning structures yields domain structures with $P_1^+$, $P_2^-$, and $P_3^+$ in a 1:2:1 ratio, $P_1^+$, $P_2^+$, and $P_3^-$ in a 1:1:2 ratio, or $P_1^-$, $P_2^+$, and $P_3^+$ in a 2:1:1 ratio. We term these domain structures as Type-II twinning structures. Phase-field modeling confirms the potential to create all domain structure variants and can be used to probe the relative energy of the different Type-I and -II twinning structures (resulting Type-I structures are shown in Figs. (1–3) and Type-II structures in Figs. (4–6) of Fig. 1 and the energies of those structures is shown in Fig. 1h). All told, there are three rotational variants of each up- and down-poled version of the two twinning structure types, and all structures are found to possess nearly identical overall energies, suggesting they are essentially degenerate stable states. This manifold degeneracy of the domain structures implies that application of the appropriate bias can enable one to push the system into any number of domain configurations and, in turn, polarization states. This raises the question whether this multi-stability can be measured and controlled at the macroscale in capacitor-based structures.

**Pulse-switching measurements.** To explore this concept, we studied the switching behavior of (111)-oriented heterostructures using a modified positive-up-negative-down (PUND) test (Supplementary Fig. 3a–c). For comparison and context, we also measured the switching behavior of (001)-oriented heterostructures. The measured switching behavior is dramatically different in the (001)-oriented (Fig. 2a) and (111)-oriented (Fig. 2b) heterostructures. For simplicity, only the positive remanent switched polarization is shown (full profiles are available, Supplementary Fig. 4). (001)-oriented heterostructures exhibit typical bipolar switching from the down- to up-poled state, where switching occurs at shorter pulse widths for higher pulse voltages. (111)-oriented heterostructures, however, exhibit multi-state

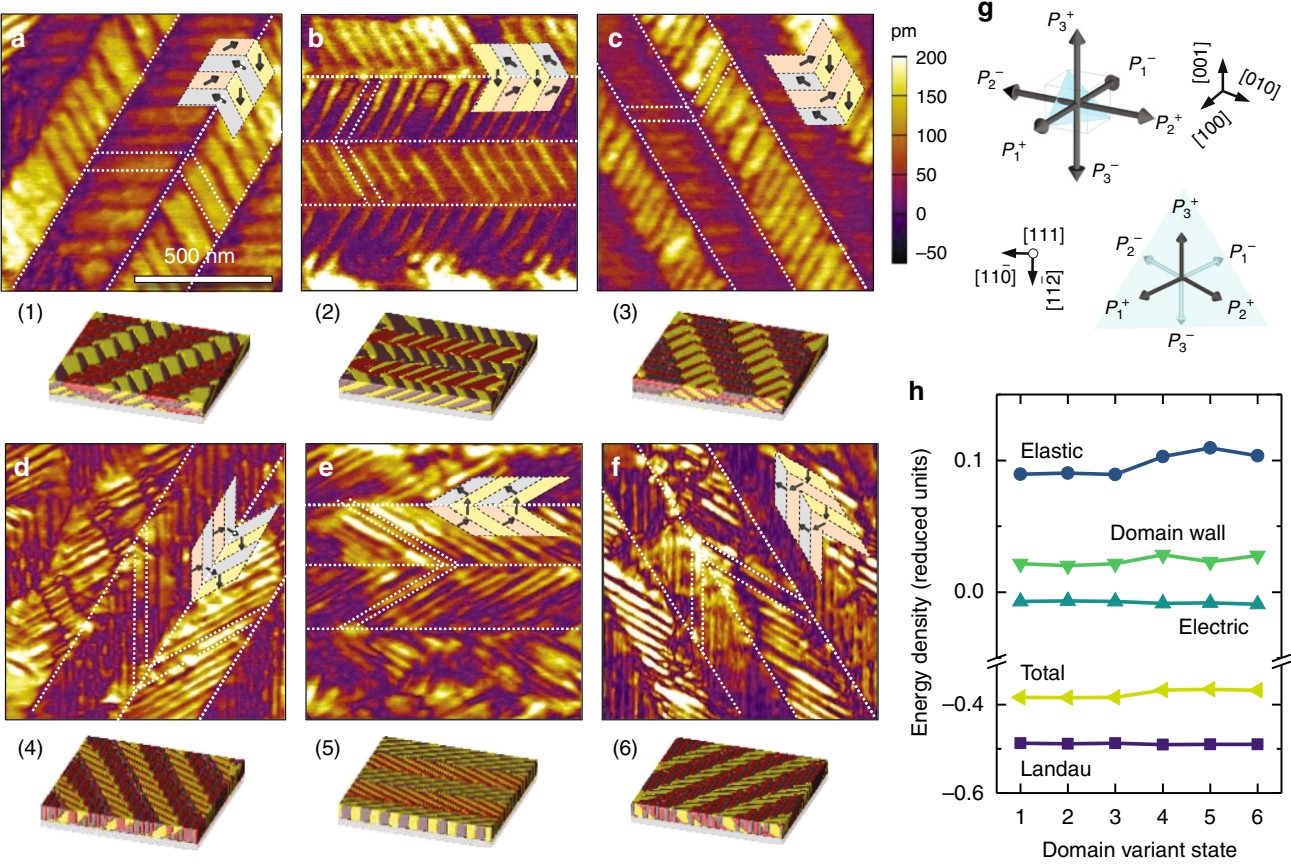

**Fig. 1** Piezoresponse force microscopy characterization of domain structures. Piezoresponse force microscopy phase images **a–c** of fully down-poled Type-I twinning structures (1/4 area $P_1^-$ and $P_3^-$ with 1/2 area of $P_2^-$ or cyclic permutations of the 3 directions) and **d–f** of Type-II twinning structures (1/4 area $P_1^-$ and $P_3^-$ with 1/2 area of $P_2^+$ or cyclic permutations of the 3 directions) which exhibit zero net out-of-plane polarization. **g** Schematic of the six possible polarization variants ($P_1^+$ and $P_1^-$ shaded in pink, $P_2^+$ and $P_2^-$ in blue, $P_3^+$ and $P_3^-$ in yellow) in (111)-oriented tetragonal ferroelectrics and their corresponding projections onto the [111]. **h** Phase-field simulation of various energetically degenerate domain structures (1)–(6) and the extracted total, Landau, electric, elastic, and domain-wall energies of the system

polarization between the 50% and fully-up-poled states with increasing pulse voltages (here 10 intermediate, 12 overall states are demonstrated; Fig. 2b). At higher voltages, the switching process evolves into a more bipolar-like process, similar to that in the (001)-oriented heterostructures. For moderate voltages, the switched polarization changes rapidly with pulse duration for (001)-oriented films, making precise specification of the polarization state difficult. By contrast, the polarization changes rapidly but then reaches a pronounced plateau for (111)-oriented films, such that a wide variety of different pulse widths yield nearly identical intermediate values of polarization.

Such different switching characteristics for the (001)- and (111)-oriented heterostructures can be further revealed from the energy landscape of polarization switching. By multiplying the switched polarization and the applied electric field, we can estimate the potential energy difference between two different states. In (001)-oriented heterostructures, a double-well potential is observed for the switching between fully down- and up-poled states, wherein the energy difference between the two states increases and the energy barrier decreases with increasing voltage (Fig. 2c). In (111)-oriented heterostructures, on the other hand, a triple-well potential with two energy barriers corresponding to switching from a fully down-poled to an intermediate-poled state and from an intermediate-poled to a fully-up-poled state are observed (Fig. 2d). The net polarization of the intermediate state evolves from zero to the fully-up-poled state with increasing voltage, which eventually leads to a change of the energy

landscape from a triple-well to a double-well structure at higher voltages. Overall, these results suggest an unusual multi-stable polarization switching process in (111)-oriented heterostructures. The set of four energetically degenerate twinning structures (fully up-polarized or down-polarized Type-I twinning structures and two versions of partially up-poled and down-poled Type-II twinning structures with zero net polarization), as well as the three rotational sub-variants of each of these, are all observed in the PFM and in phase-field simulation studies. This rich set of twinning structures certainly provides a structural foundation for the observed multi-state switching phenomena, yet these observations alone cannot immediately account for the much greater number of states obtained in the pulse-switching measurement.

**Molecular dynamics simulations**. To answer the question about how this system accommodates the multi-state switching phenomenon, MD simulations were performed to examine the switching process in realistic domain structures under varying applied electric fields along the [111] of a $PbTiO_3$ supercell (Supplementary Fig. 5). We expect that the domain switching mechanisms simulated with $PbTiO_3$ are comparable to those in $PbZr_{0.2}Ti_{0.8}O_3$ given the structural similarity between these two ferroelectric materials. Here MD simulations reveal a bipolar-like switching process at high fields and a multi-step switching process, characterized by a polarization plateau, at low fields

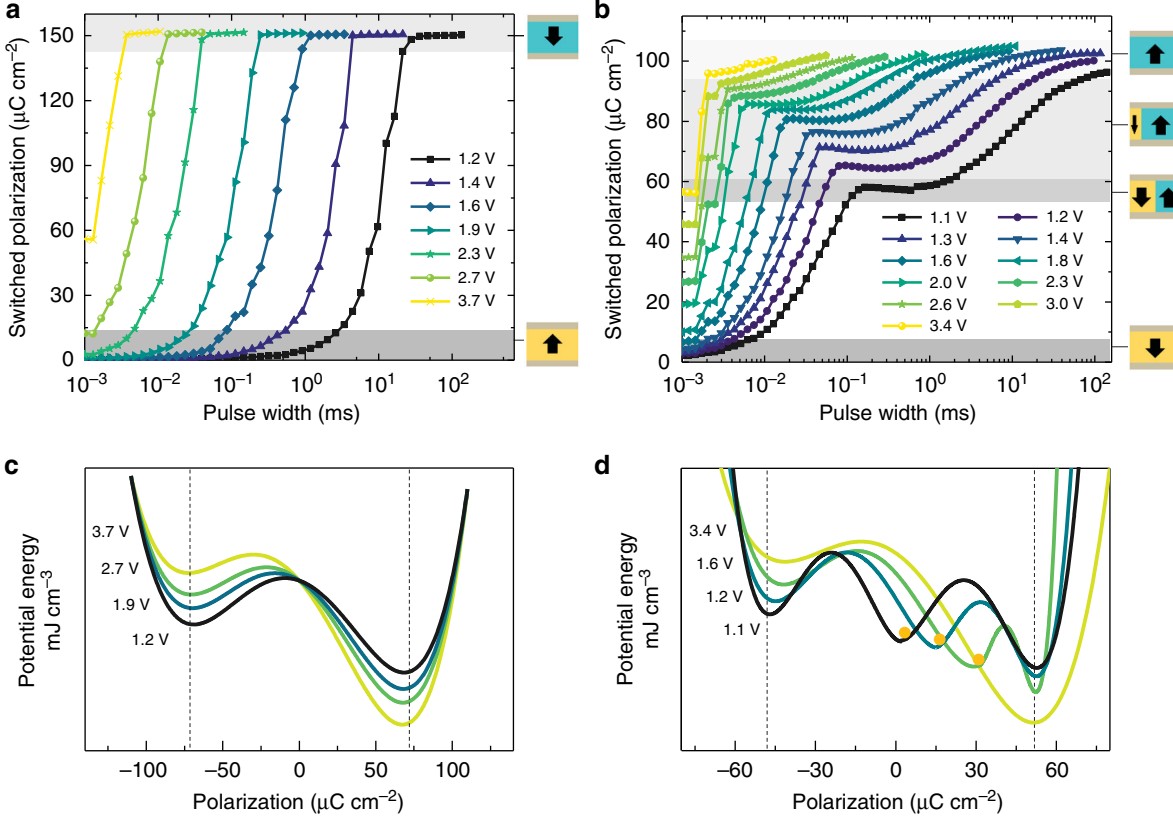

**Fig. 2** Pulse-switching measurements. Switched polarization measured as a function of pulse width for different pulse voltages at a pulse delay of 1 ms in (**a**) (001)-oriented heterostructures, which exhibit typical bipolar switching from the down- to up-poled state, and (**b**) (111)-oriented heterostructures, which exhibit multi-state polarization between the 50% and fully-up-poled states with increasing pulse voltages. **c** The estimated double-well energy landscape in (001)-oriented and (**d**) the triple-well energy landscape in (111)-oriented heterostructures

(Fig. 3a), which are consistent with the pulsed-switching result. The domain structure evolution was also studied and is mapped in a spherical coordinate-based color scheme (Fig. 3b, c). For the case of bipolar-like switching at high fields ($E_5$, Fig. 3d–i), the fully-poled domain structure (consistent with the Type-I twinning structure observed in PFM and phase-field simulation studies, Fig. 3d) undergoes a series of coordinated (nearly simultaneous) 90° switching events including $P_3^- \rightarrow P_2^+$, $P_1^- \rightarrow P_3^+$, $P_2^- \rightarrow P_3^+$, and $P_3^- \rightarrow P_1^+$ (Fig. 3g, h) without a substantial change in the overall twinning structure (albeit local polar vectors are rotated by 90°). In the case of the multi-step switching at low fields ($E_1$, Fig. 3d, j–n), the polarization plateau (Fig. 3a) corresponds to the formation of a new domain configuration similar to the Type-II twinning structures observed in PFM and phase-field simulation studies (Fig. 3m), with one up-poled band ($P_2^+/P_3^+$) and one down-poled band ($P_1^-/P_3^-$) (with 50% reversal of the out-of-plane polarization) resulting from three 90° switching events: $P_3^- \rightarrow P_2^+$ and $P_2^- \rightarrow P_3^+$ (Fig. 3k, l) wherein the out-of-plane component of polarization is reversed and $P_3^- \rightarrow P_1^-$ (Fig. 3m) wherein the out-of-plane component of polarization is unchanged. The volume fraction of the down-poled band in the Type-II twinning structure is further gradually reduced with the continued application of the field and eventually transforms to a fully up-poled Type-I twinning structure identical to that achieved in the high-field switching (Fig. 3n).

The MD simulations provide further insights into the mechanism underlying the realization of this multi-state polarization evolution: first, consistent with prior work, 90° switching is favored in (111)-oriented heterostructures, regardless of the field strength, due to a lower-energy barrier[26]; this is further verified by phase-

field simulations (Supplementary Fig. 6). Second, (111)-oriented heterostructures possess two switching pathways, including direct bipolar-like switching at high fields (transiting from down-poled Type-I → up-poled Type-I through coordinated single-step 90° switching events) and multi-step 90° switching at low fields (transiting from down-poled Type-I → half down-poled Type-II → up-poled Type-I). Third, The tunable and stable intermediate polarization states are created by varying the proportion of the two switching pathways. Specifically, at low fields, only multi-step 90° switching is active, giving rise to a 50% poled Type-II twinning structure. At intermediate fields, both switching pathways are active simultaneously, and the kinetic balance of these two switching pathways leads to a series of intermediate polarization states with different volume fractions of Type-I and -II structures (and thus different stable out-of-plane polarization states). At high fields, bipolar-like, direct 90° switching is active, giving rise to the fully up-poled Type-I twinning structure.

**Visualizing the multi-state polarization.** The ability to manipulate the fraction of two distinct twinning structures was experimentally confirmed in capacitor-based structures (Supplementary Figs. 7 and 8) which were pre-poled into different states including: fully down-poled [(b) in Fig. 4a], two intermediate [(c), (d) in Fig. 4a], and fully up-poled [(e) in Fig. 4a] states. Following poling, the top contacts were removed and the underlying domain structures mapped (Methods). PFM analysis reveals the presence of two distinct domain configurations, namely, Type-I and -II twinning structures. To better illustrate this coexistence of the two twinning structures, we map regions of Type-I (yellow-orange regions in the PFM images,

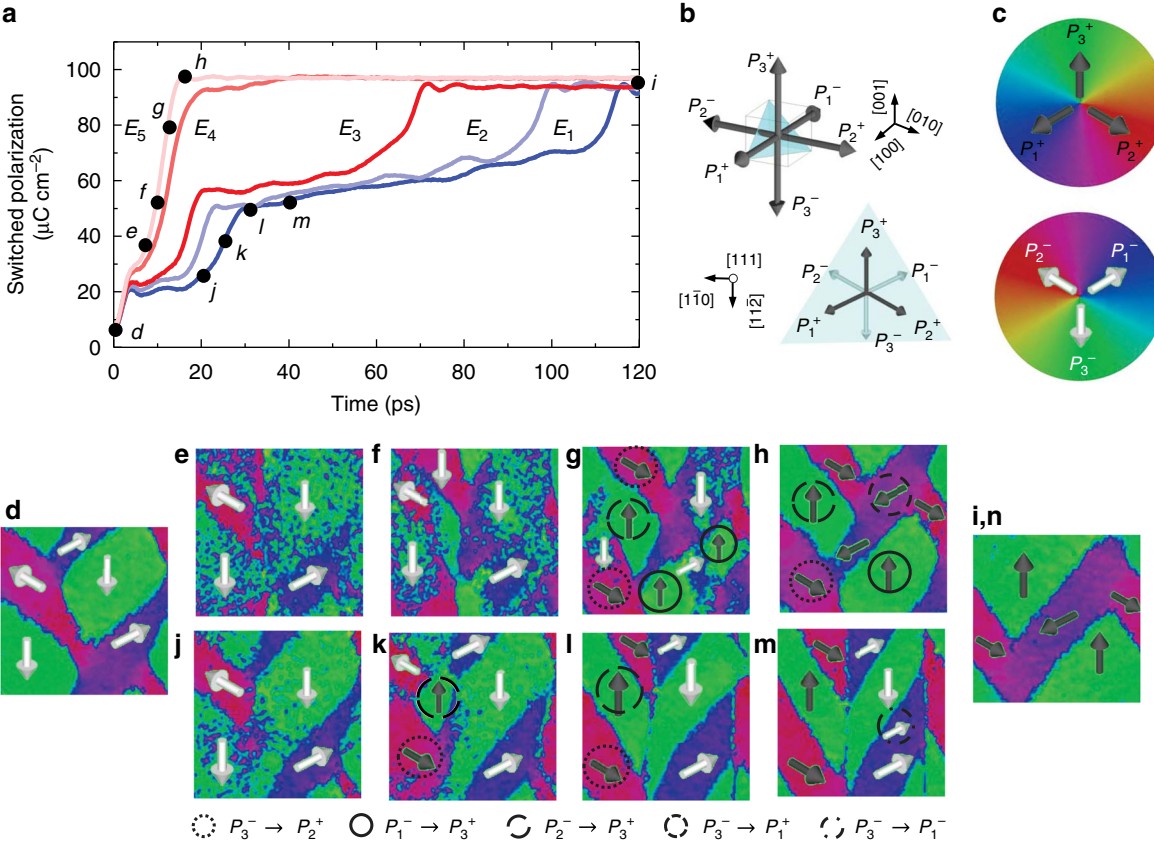

**Fig. 3** Molecular-dynamics simulations. **a** Evolution of switched polarization in response to electric field applied along the [111] for different field magnitudes ($E_1 < E_2 < E_3 < E_4 < E_5$). **b** Schematics of the six possible polarization variants in (111)-oriented tetragonal ferroelectrics and their corresponding projections on the (111). To guide the visualization, the polarization vectors with positive [111] components are colored in black and those with negative [111] components are colored in white. **c** Spherical coordinate-based color scheme used to represent the up-poled polarization variants (top panel) and the down-poled polarization variants (bottom panel). Simulated domain evolution under (**d-i**), high ($E = E_5$) and (**d, j-n**), low ($E = E_1$) fields revealing two distinct switching pathways. Under low fields, the domains experience substantial changes in the twinning structures, characterized by an intermediate state with partially-switched polarization (**k-m**). Under high fields, three coordinated 90° switching events **g**, enable rapid ferroelectric switching without significant change in twinning structure orientation

Fig. 4b–e) and Type-II (blue regions in the PFM images, Fig. 4b–e) structures in the PFM images. As expected, the initial poled state exhibits a fully down-poled Type-I twinning structure with uniform contrast in vertical PFM images (Fig. 4b and Supplementary Fig. 9a, b). Type-II twinning appears in the first intermediate state, resulting in an almost 50% reversed contrast in vertical PFM (Fig. 4c and Supplementary Fig. 9c, d). Furthermore, a mixture of Type-I and -II twinning was observed in the second intermediate state with the Type-I twinning becoming fully up-poled, leading to a reversal of the vertical contrast (Fig. 4d and Supplementary Fig. 9e, f). Finally, the system transforms to the fully up-poled Type-I twinning, exhibiting a uniformly reversed vertical contrast (Fig. 4e and Supplementary Fig. 9g, h). These observations verify the mechanism responsible for the multi-state polarization suggested by the MD simulations, where the voltage-based control of two competing switching pathways (Fig. 4f–h) enables the manipulation of the volume fraction of Type-I and -II twinning structures.

Such control of the switching pathways is enabled by the coordinated ferroelastic switching that occurs in these structures. Combining PFM and MD results, a complete picture can be generated to understand the switching mechanisms active in the (111)-oriented heterostructures. First, for multi-step switching (Fig. 4f), two types of 90° switching events are identified: sign-changing switching wherein there is a change in the out-of-plane component of polarization (e.g., $P_3^- \rightarrow P_1^+$) and sign-conserving

switching wherein there is no change in the out-of-plane component of polarization (e.g., $P_1^- \rightarrow P_3^-$ and $P_1^+ \rightarrow P_3^+$). Note that for simplicity, here we only show the switching process in the $P_1^-/P_3^-$ band, but a similar process also proceeds in the $P_1^-/P_2^-$ band including sign-changing switching $P_2^- \rightarrow P_1^+$ and sign-conserving switching $P_1^- \rightarrow P_2^-$ and $P_1^+ \rightarrow P_2^+$. For bipolar-like switching (Fig. 4g), only sign-changing switching occurs ($P_1^- \rightarrow P_3^+$ and $P_3^- \rightarrow P_1^+$). Again, the similar process also proceeds in the $P_1^-/P_2^-$ band including sign-changing switching $P_1^- \rightarrow P_2^+$ and $P_2^- \rightarrow P_1^+$. Energetically, sign-changing switching is favored thermodynamically by an electro-static energy gain $-EP$, but is associated with an elastic-energy barrier $\Delta E_e^1$, while sign-conserving switching is associated only with an elastic-energy barrier $\Delta E_e^2$. Moreover, $\Delta E_e^1 >> \Delta E_e^2$ because the transition state of sign-changing switching puts the polarization vector fully within the (111) and thus subject to stronger clamping effects, whereas the transient state of sign-conserving switching has a smaller in-plane component of polarization (Fig. 4h, i). Thus, at low fields (where the elastic energy dominates), as sign-changing switching occurs first in one domain, the neighboring domain compensates the strain via a sign-conserving switching event, which is much more rapid kinetically due to the lower elastic-energy barrier. This helps rationalize the formation of the 50% poled Type-II twinning structure with half of the domains up-poled (resulting from the

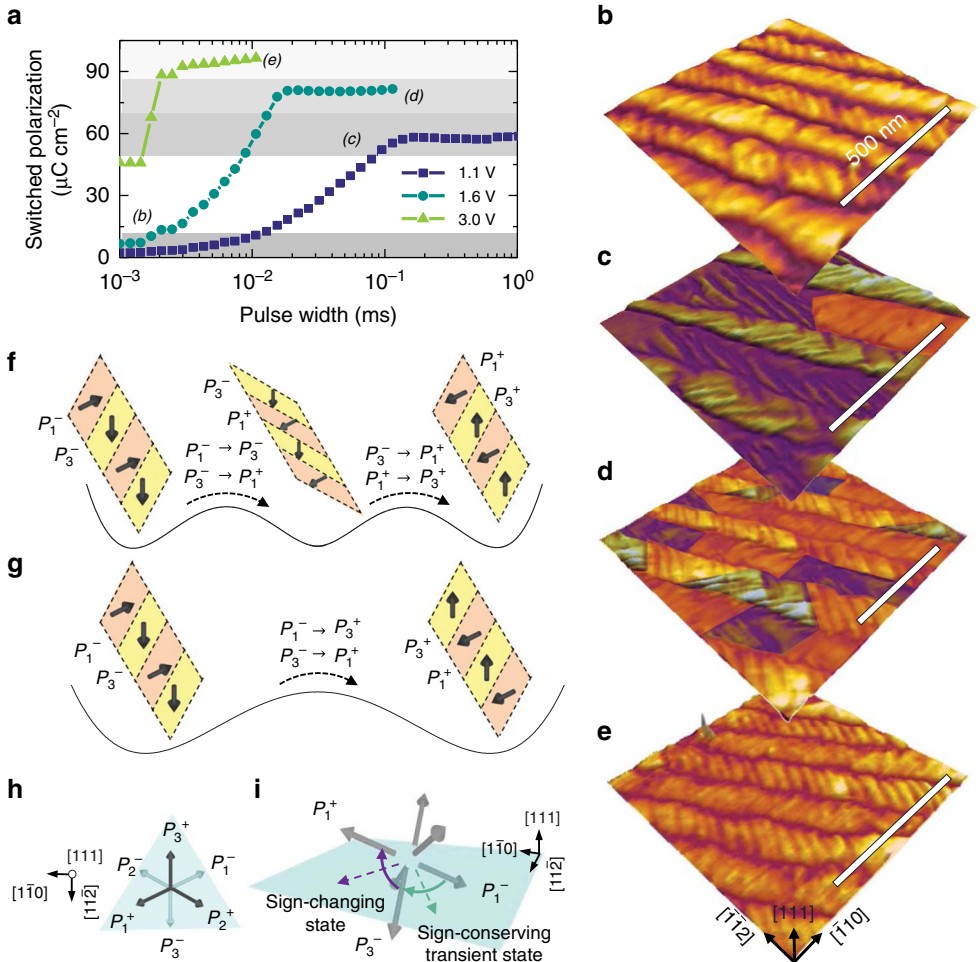

**Fig. 4** Visualizing domain structures for multi-state polarization. **a** Multiple polarization states were pre-poled at pulse voltages of 1.1, 1.6, and 3 V. Piezoresponse force microscope imaging of domain structures and the evolution in the volume fraction of the two types of twinning structures in polarization states for the (**b**) initial, fully up-poled state, the intermediate states poled at (**c**) 1.1 V and (**d**) 1.6 V, as well as **e**, the final fully down-poled state poled at 3.0 V. In these figures, the Type-I twinning structures are shaded yellow-orange and the Type-II twinning structures are shaded blue. Schematics of the **f**, multi-step switching pathway at low fields which transitions from fully down-poled Type-I structures to half down-poled Type-II structures to fully up-poled Type-I structures and (**g**) direct bipolar-like switching pathway at high fields which transitions from fully down-poled Type-I structures to fully up-poled Type-I structures. **h** Plan-view schematic of the six possible polarization variants. **i**, Schematic illustrating the sign-changing and –conserving switching events with the transient states noted

sign-changing switching) and half of the domains down-poled (resulting from the sign-conserving switching). The net-zero out-of-plane polarization of the Type-II twinning structure also helps stabilize this structure, due to the absence of a depolarization field. At high fields (where the electrostatic energy dominates), the system thermodynamically prefers to proceed through more sign-changing switching events that have a larger electrostatic-energy gain. This explains the bipolar-like switching pathway accomplished through coordinated single-step, sign-changing switching without substantial changes in the twinning structures. When the electric field is intermediate, such that the electrostatic energy $EP$ is comparable to $\Delta E_e{}^1 - \Delta E_e{}^2$ (Bell-Evans-Polanyi principle[28,29]), both switching pathways are comparably fast, producing a mixture of Type-I and -II twinning structures and different intermediate polarization states.

**Properties of multi-state polarization**. With this understanding of the switching mechanism underlying the multi-state polarization of the (111)-oriented heterostructures, we proceed to study the performance potential of these intermediate

states including their retention, repeatability, and endurance in capacitor-based measurements. As was done before, we compare the properties of the multi-state polarization in the (111)-oriented heterostructures with those of the partially-switched states in (001)-oriented heterostructures. Here, the stability of the various states in the (001)-oriented (Supplementary Fig. 10a) and (111)-oriented (Fig. 5a) heterostructures were probed using a retention-pulse sequence (Supplementary Fig. 3d). In the (001)-oriented heterostructures, the fully down- and up-poled states are the only stable states; all others decay with time (Supplementary Fig. 10a). All states, on the other hand, in the (111)-oriented heterostructures are stable, showing <5% change in polarization even after about 8 h (Fig. 5a and Supplementary Fig. 11). In terms of deterministic repeatability of reaching a given state, the (001)- and (111)-oriented heterostructures also show remarkable differences. Taking the average of ten pulse measurements, the (001)-oriented heterostructures exhibit poor repeatability, with error bars in the partially-switched regime >50% of the switched polarization value, due to the stochastic nature of switching and extrinsically mediated mechanisms to

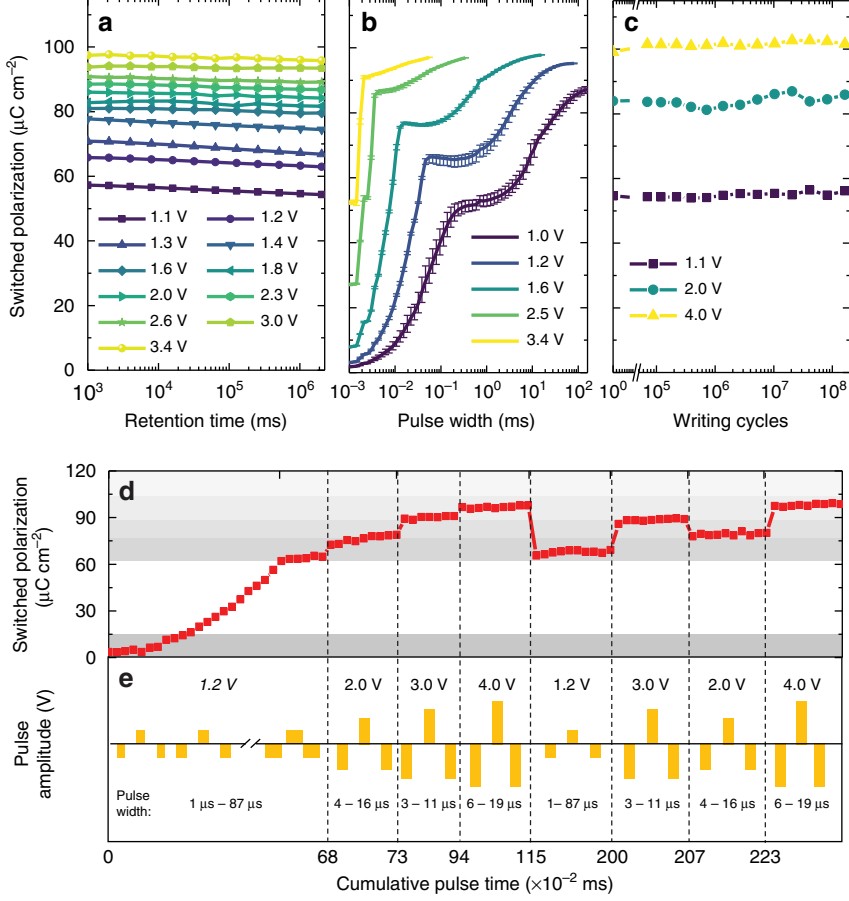

**Fig. 5** Properties of multi-state polarization in heterostructures. **a** Retention measurements probing the time stability of the intermediate-polarization states. **b** Repeatability measurements wherein the error bars were generated after repeating each pulse-switching measurement 10 times. **c** Endurance measurements of the switched polarization as a function of writing cycles. **d** Demonstration of on-demand switching ability including ascending and arbitrary access to any desired polarization state by simultaneously controlling the pulse duration and amplitude. **e** The schematic illustration of pulse series used in **d**

pin domain walls after switching (e.g., pinning by random defects, see Methods and Supplementary Fig. 10b). The (111)-oriented heterostructures, however, reveal deterministic accessibility, with repeated writing taking the material to the same polarization state to within <5% polarization variation (Fig. 5b). The endurance of each polarization state was also explored using a fatigue-pulse sequence (Methods and Supplementary Fig. 3e). The (001)-oriented heterostructures exhibit poor fatigue properties, as different partially-switched states converge to a similar value and provide diminished distinguishability after repeated cycling (Methods and Supplementary Fig. 10c). By contrast, each of the polarization states in the (111)-oriented heterostructures exhibit robust distinguishability even after $10^8$ writing cycles (Fig. 5c). We also studied how to control these polarization states such that they can be deterministically placed into a desired value in an on-demand fashion. Here, we achieved such control of polarization switching in the (111)-oriented heterostructures by simultaneously controlling the pulse voltage and width (Fig. 5d, e). Overall, these results suggest that (111)-oriented heterostructures exhibit a number of features that are important for robust multi-state operation, including deterministic stimuli-driven access to precisely defined and distinguishable states, non-volatility and retention of those states, and endurance of those states to repeated modification without loss of distinguishability. This stands in stark contrast to the use of classical bi-stable switching (e.g., in (001)-oriented

heterostructures) where it is difficult, if not impossible, to achieve such desired function.

In summary, we demonstrate an approach to move beyond the inherent bi-stability of ferroelectrics and realize deterministic multi-state polarization. This multi-state polarization is made possible by the fact that the material can transition through multiple, kinetically mediated switching pathways. The control of switching pathways is enabled by the competition of elastic and electrostatic energies which favors different types of ferroelastic switching that can occur. The switching process is kinetically driven via multi-step switching at low fields and thermodynamically driven via bipolar switching at high fields. At intermediate fields, both switching pathways are comparably fast, producing a mixture of Type-I and -II twinning structures and different intermediate polarization states. As a result, it is possible to create tunable intermediate states in ferroelectrics without introducing defect-induced pinning or other extrinsic factors, while addressing the requirements of deterministic and repeatable production of multi-state polarization, which holds promise for non-volatile multi-state applications.

## Methods

**Thin-film growth**. 80 nm La$_{0.7}$Sr$_{0.3}$MnO$_3$/100 nm PbZr$_{0.2}$Ti$_{0.8}$O$_3$/20 nm La$_{0.7}$Sr$_{0.3}$MnO$_3$ heterostructures were grown on (001)-oriented and (111)-oriented, single-crystalline SrTiO$_3$ substrates by pulsed-laser deposition. The growth of all film layers was carried out in a dynamic oxygen pressure of 200 mTorr, at a growth temperature of 650 °C, and a laser fluence and repetition rate of 1.0 J cm$^{-2}$ and 3 Hz, respectively. Following growth, the samples were cooled to room temperature at a cooling rate of 5 °C min$^{-1}$. under at a static oxygen pressure of 760 Torr.

**Ferroelectric hysteresis loop measurements**. Hysteresis loop measurements were performed on $PbZr_{0.2}Ti_{0.8}O_3$ capacitors with symmetric $La_{0.7}Sr_{0.3}MnO_3$ electrodes using a Precision Multiferroic Tester (Radiant Technologies, Inc.). The circular top electrodes (diameter 12.5 μm) were fabricated using a wet etching method (Supplementary Fig. 7). First, photoresist was patterned on the as-grown heterostructures using photolithography which only covers the circular electrode regions (Supplementary Fig. 7b). Using dilute $H_3PO_4$ acid (dilution ratio 1 part acid : 5 parts water), the uncovered $La_{0.7}Sr_{0.3}MnO_3$ was etched away within 30 s, leaving circular $La_{0.7}Sr_{0.3}MnO_3$ contacts covered by the photoresist. Subsequently, the photoresist was removed with acetone (Supplementary Fig. 7c).

**Pulsed-switching measurements**. We performed pulsed-switching measurements using a modified PUND pulse sequence on the symmetric capacitor structures using a Precision Multiferroic Tester (Radiant Technologies, Inc.). In a conventional PUND measurement, the ferroelectric capacitor is subjected to the pulse sequence including one preset pulse and four reading pulses (Supplementary Fig. 3a). The first preset pulse is used to pre-pole the capacitor to the $-V$ direction; no measurement will be performed as a result of this pulse. The second pulse switches the capacitor to the $+V$ direction and measures the amount of switched polarization. The third pulse is a twin pulse of the second, which is used to read the non-remanent part of polarization that dissipates during the delay time after the second pulse. The fourth and fifth pulses mirror the second and third and are used to switch the capacitor to the $-V$ direction and read both the switched and non-switched polarization, respectively. There are eight measured parameters generated by the PUND measurement, namely $P_i^*$ and $P_i^r$ ($i = 2, 3, 4, 5$, asterisk (*) represents the sum of remanent and non-remanent polarization components, r represents the remanent polarization component only), where $P_i^*$ is measured at the end of each reading pulse when the voltage is on, and $P_i^r$ is measured after a certain amount of delay time when the voltage is off. These eight parameters can be measured as a function of pulse time and voltage by varying certain pulse parameters of the basic waveform.

Since we focused on the measurement of switched remanent polarization $P_i^r$ read by the second and the fourth pulse instead of non-switched polarizations read by the third and the fifth pulse, we used a modified PUND pulse sequence wherein the third and fifth pulses are removed from the original PUND waveform (Supplementary Fig. 3b). The pulse-switching measurement was performed as a function of pulse width for a certain pulse voltage (Supplementary Fig. 3c). Each measurement cycle consists of three pulses with equal time widths. The pulse voltage is kept constant while the pulse width is varied incrementally for different cycles. For simplicity, only the positive remanent switched polarization is shown (Fig. 2a, b), which is measured at 0 V after a delay of 1 s from the fall of the positive reading pulse. The full switching profile is provided where both the positive and negative $P_i^r$ ($i = 2, 3$) are shown here for (001)-oriented and (111)-oriented films, respectively (Supplementary Fig. 4).

**Retention measurements**. The retention measurement is performed using a Precision Multiferroic Tester (Radiant Technologies, Inc.). The retention measurement pulse sequence is provided (Supplementary Fig. 3d). In this measurement, the material is first switched into a certain polarization state (e.g., partially-switched states and intermediate states in (001)-oriented and (111)-oriented heterostructures, respectively) and then the system is held at 0 V for a certain amount of retention time. After this waiting period, a pair of reading pulses is applied to read the variation in both the positive and negative switched polarization. Furthermore, the retention time can be varied to allow the measurement of polarization variations as a function of retention time.

**Fatigue measurement**. The fatigue measurement is performed using a Precision Multiferroic Tester (Radiant Technologies, Inc.). The adapted fatigue measurement pulse sequence is provided (Supplementary Fig. 3e). In this measurement, the fatigue cycles were applied to the capacitor to switch the capacitor as many as $1.64 \times 10^8$ cycles. In order to reproduce the writing of various multi-state polarization values, the fatigue cycles were set to include a high negative voltage pulse which switches the capacitor into a fully up-poled state followed by a smaller positive voltage pulse which switches the capacitor from the fully up-poled state into one of many multi-state polarization values. The negative pulse voltage was $-5.0$ V for (001)-oriented and $-4.0$ V for (111)-oriented heterostructures, respectively, and the positive pulse voltage varies from 1.8 to 5.0 V and 1.1 to 4.0 V for the (001)-oriented and (111)-oriented heterostructures, respectively. The pulse width used in these studies was set to be 0.005 ms. The fatigue cycles were interrupted regularly by a pair of read pulses which monitors the variation of remanent polarization as a function of writing cycles.

**Tuning intermediate states with voltage pulse magnitude and width**. In this measurement, the intermediate states can be tuned in an ascending (descending) or an arbitrary manner (Fig. 5e, f). For instance, to access intermediate states in an ascending manner, we first use $-1.2$ V, 1–87 μs pulses to switch the polarization into a 60% up-poled state. If we then change to $-2.0$ V, 4–16 μs pulses, we can further switch the 60% up-poled state into a 75% up-poled state. Similarly, we can further tune the 75% up-poled state into a 90% and finally into a 100% up-poled state by changing to $-3.0$ V, 3–11 μs and $-4.0$ V, 6–19 μs pulses, respectively.

Similarly, to demonstrate that we can access intermediate states in an arbitrary manner, we again use $-1.2$ V, 1–87 μs pulses to switch the polarization into a 60% up-poled state. If we then change to $-3.0$ V, 3–11 μs pulses, we can further switch the 60% up-poled state into a 90% up-poled state. Similarly, we can further tune the 90% up-poled state into a 75% and finally into a 100% up-poled state by changing to $-2.0$ V, 4–16 μs and $-4.0$ V, 6–19 μs pulses, respectively.

**Crystal and domain structure characterization**. X-ray $\theta$-2$\theta$ scans were obtained by high-resolution X-ray diffraction (XPert MRD Pro equipped with a PIXcel detector, Panalytical). The PFM studies were carried out in areas of pre-poled capacitors by a MFP-3D AFM (Asylum Research) using Ir/Pt-coated conductive tips (Nanosensor, PointProbe® Plus Electrostatic Force Microscopy, force constant ≈2.8 N m$^{-1}$). In order to image the domain structure under the pre-poled capacitors, first we pre-poled multiple capacitors into different polarization states and then coated a layer of inversely patterned photoresist using photolithography that only covers the $PbZr_{0.2}Ti_{0.8}O_3$ film region (Supplementary Fig. 7d). The uncovered top $La_{0.7}Sr_{0.3}MnO_3$ electrodes could be readily etched away using dilute $H_3PO_4$ acid (dilution ratio 1 part acid:5 parts water) (Supplementary Fig. 7e). The inversely patterned photoresist was left on the film surface serving as a marker that aids the accurate location and imaging for regions that were under the pre-poled capacitors. Here we also provide PFM results of four states discussed in Fig. 4 that were scanned in the entire area of capacitors (Supplementary Fig. 8).

**Molecular-dynamics simulations**. The molecular dynamics simulations were performed on a (111)-oriented $28\sqrt{6} \times 50\sqrt{2} \times 16\sqrt{3}$ supercell of $PbTiO_3$ (672,000 atoms, Supplementary Fig. 5) with periodic boundary conditions and a bond-valence-based interatomic potential parameterized from first-principles[30]. We expect that the domain switching mechanisms simulated with $PbTiO_3$ are comparable to those in $PbZr_{0.2}Ti_{0.8}O_3$ given the structural similarity between these two ferroelectric materials. The supercell has the Cartesian axes aligned along crystallographic axes [11$\bar{2}$], [1$\bar{1}$0], and [111], respectively. We first equilibrate the supercell by running constant-pressure, constant-temperature simulations using a time step of 1 fs. The temperature is controlled via the Nosé-Hoover thermostat, and the pressure is maintained at 1 atm via the Parrinello-Rahman barostat implemented in Large-scale Atomic/Molecular Massively Parallel Simulator (LAMMPS). Once equilibrium is reached, a stable configuration is used as a starting point for runs in the presence of external electric fields. To capture the mechanical clamping effect in epitaxial thin-film experiments, the supercell dimensions along [11$\bar{2}$] and [1$\bar{1}$0] are fixed when applying the [111] electric field, while the [111] dimension is free to relax.

**Domain color scheme for the MD simulations**. We color the domains based on the direction of local polarization within each unit cell. The instantaneous local polarization, $\mathbf{P}_u^m(t)$, centered at a Ti atom within the unit cell $m$ is

$$\mathbf{P}_u^m(t) = \frac{1}{V_u}\left(\frac{1}{8}\mathbf{Z}_{Pb}^* \cdot \sum_{i=1}^{8}\mathbf{r}_{Pb,i}(t) + \mathbf{Z}_{Ti}^* \cdot \mathbf{r}_{Ti}(t) + \frac{1}{2}\mathbf{Z}_O^* \cdot \sum_{i=1}^{6}\mathbf{r}_{O,i}(t)\right) \qquad (1)$$

where $V_u$ is the volume of a unit cell, $\mathbf{Z}_{Pb}^*$, $\mathbf{Z}_{Ti}^*$, and $\mathbf{Z}_O^*$ are the Born effective charge tensors of Pb, Ti, and O atoms; $\mathbf{r}_{Pb,i}(t)$, $\mathbf{r}_{Ti,i}(t)$ and $\mathbf{r}_{O,i}(t)$ are instantaneous atomic positions of Pb, Ti, and O atoms in unit cell $m$ obtained from MD simulations. The above equation is essentially the polarization resulting from the Ti-centered local dipole moment formed by the Ti and its nearest eight Pb atoms and six O atoms. For each local polarization vector, we then calculate the angle ($\theta$) between its projection on [111] plane and [11$\bar{2}$] axis and the polarization component ($P_{[111]}$) along the [111] axis. We introduce two HSV (Hue, Saturation, Value) color wheels, one for positive $P_{[111]}$ and one for negative $P_{[111]}$, with one wheel rotated by 180° compared to the other wheel. We then color the local polarization based on the value of $\theta$ by picking one color wheel depending on the sign of $P_{[111]}$. This color scheme will make $P_i^{\pm}$ ($i = 1, 2, 3$) of the same color, similar to experimental color scheme. (Fig. 3b, c). The simulated domain pattern (viewed from the [111]; Fig. 3d, j) consists of nano-twinned domain bands, each containing two of the three degenerate polarization variants ($P_1^-$, $P_2^-$, and $P_3^-$), closely resembling the domain structures in (111)-oriented heterostructures observed experimentally.

**Phase-field modeling**. Phase-field modeling was performed to study local switching events and domain structure variants in (001)-oriented and (111)-oriented heterostructures. Here two coordinate systems are used including $\mathbf{x}$ ($x_1$, $x_2$, $x_3$) with $x_1$, $x_2$, and $x_3$ along [100], [010], and [001] and $\mathbf{x}'$ ($x_1'$, $x_2'$, $x_3'$) with $x_1'$, $x_2'$, and $x_3'$ along [01$\bar{1}$], [$\bar{2}$11], and [111] for the (001)-oriented and (111)-oriented heterostructures, respectively. The polarization $\mathbf{P}'$ in the $\mathbf{x}'$ coordinate system can be related to the total free energy $F$ through time-dependent Landau-Ginzburg (TDGL) equation[31,32],

$$\frac{\partial P_i'(\mathbf{r}, t)}{\partial t} = -L\frac{\delta F}{\delta P_i'(\mathbf{r}, t)}, \qquad (2)$$

where $L$ is the kinetic coefficient. The total free energy $F$ includes bulk, elastic,

electric, and gradient energies,

$$F = \iiint_V \left( f_{\text{bulk}} + f_{\text{elastic}} + f_{\text{electric}} + f_{\text{grad}} \right) dV, \tag{3}$$

where $V$ represents the volume of ferroelectric thin films. The bulk free energy density for $PbZr_{0.2}Ti_{0.8}O_3$ is described by a sixth-order Landau–Devonshire polynomial[33],

$$
\begin{aligned}
f_{\text{bulk}} = {} & \alpha_1 \left( P_1^2 + P_2^2 + P_3^2 \right) + \alpha_{11} \left( P_1^4 + P_2^4 + P_3^4 \right) \\
& + \alpha_{12} \left( P_1^2 P_2^2 + P_1^2 P_3^2 + P_2^2 P_3^2 \right) \\
& + \alpha_{112} \left[ P_1^4 \left( P_2^2 + P_3^2 \right) + P_2^4 \left( P_1^2 + P_3^2 \right) \right. \\
& \left. + P_3^4 \left( P_1^2 + P_2^2 \right) \right] \\
& + \alpha_{111} \left( P_1^6 + P_2^6 + P_3^6 \right) + \alpha_{123} P_1^2 P_2^2 P_3^2,
\end{aligned}
\tag{3}
$$

where all of the coefficients are independent of temperature except $\alpha_1$. The elastic energy density is given by

$$
\begin{aligned}
f_{\text{elastic}} &= \tfrac{1}{2} c_{ijkl} e_{ij}(\mathbf{r}) e_{kl}(\mathbf{r}) \\
&= \tfrac{1}{2} c_{ijkl} \left( \varepsilon_{ij}(\mathbf{r}) - \varepsilon_{ij}^0(\mathbf{r}) \right) \left( \varepsilon_{kl}(\mathbf{r}) - \varepsilon_{kl}^0(\mathbf{r}) \right),
\end{aligned}
\tag{4}
$$

where $c_{ijkl}$, $e_{ij}(\mathbf{r})$, $\varepsilon_{ij}(\mathbf{r})$, $\varepsilon_{ij}^0(\mathbf{r})$ are the elastic stiffness tensor, elastic strain, total strain, and spontaneous strain, respectively. The spontaneous strain $\varepsilon_{ij}^0(\mathbf{r})$ can be expressed using electrostrictive coefficients and polarizations, i.e., $\varepsilon_{ij}^0(\mathbf{r}) = Q_{ijkl} P_k(\mathbf{r}) P_l(\mathbf{r})$, $i, j, k, l = 1, 2, 3$, where the summation convention for the repeated indices is employed. The total strain $\varepsilon_{ij}(\mathbf{r})$ can be expressed as a sum of the homogeneous and heterogeneous strain, i.e., $\varepsilon_{ij}(\mathbf{r}) = \bar{\varepsilon}_{ij} + \delta \varepsilon_{ij}(\mathbf{r})$, based on Khachaturyan's elastic theory[34]. The homogeneous strain is defined in such a way so that $\int \delta \varepsilon_{ij}(\mathbf{r}) dV = 0$, which represents the macroscopic shape change of a system generated due to the formation of domain structures and can be determined from the mismatch between thin films and substrates. The heterogeneous strain does not change the macroscopic shape of a system and can be calculated from $\delta \varepsilon_{ij}(\mathbf{r}) = \frac{1}{2} \left[ \partial u_i(\mathbf{r}) / \partial r_j + \partial u_j(\mathbf{r}) / \partial r_i \right]$, where $u_i(\mathbf{r})$ is the position-dependent displacement. The equilibrium elastic strain $e_{ij}(\mathbf{r})$ can be obtained by solving the mechanical equilibrium equation $\sigma_{ij,j} = 0$.

The electrostatic energy density of given domain structures can be calculated by

$$
\begin{aligned}
f_{\text{electric}} &= - \int_0^{\mathbf{E}} \mathbf{D}(\mathbf{r}, \mathbf{E}) \cdot d\mathbf{E} = -\mathbf{P}(\mathbf{r}) \cdot \mathbf{E} - \tfrac{1}{2} \varepsilon_0 \kappa^b \mathbf{E}^2 \\
&= -P_i(\mathbf{r}) E_i(\mathbf{r}) - \tfrac{1}{2} \varepsilon_0 \kappa_{ij}^b E_i(\mathbf{r}) E_j(\mathbf{r}),
\end{aligned}
\tag{5}
$$

where $\kappa_{ij}^b$ is the background dielectric constant tensor. The gradient energy density in an anisotropic system can be expressed as

$$f_{\text{grad}} = \tfrac{1}{2} G_{ijkl} P_{i,j} P_{k,l}, \tag{6}$$

where $G_{ijkl}$ is the gradient energy coefficient and $P_{i,j} = \partial P_i / \partial r_j$. For an isotropic system, Eq. (6) reduces to[35]:

$$
\begin{aligned}
f_{\text{grad}} = \tfrac{1}{2} G_{11} \big( & P_{1,1}^2 + P_{2,2}^2 + P_{3,3}^2 + P_{1,2}^2 \\
& + P_{2,1}^2 + P_{1,3}^2 + P_{3,1}^2 + P_{2,3}^2 + P_{3,2}^2 \big),
\end{aligned}
\tag{7}
$$

where $G_{ij}$ is related to $G_{ijkl}$ by the Voigt's notation, and $G_{12} = 0$, $G_{11} = 2G_{44} = 2G_{44}'$ in an isotropic system.

In addition, the polarization components $P_j$ in coordinate system $\mathbf{x}$ can be transformed to $P_i'$ in coordinate system $\mathbf{x}'$ using transformation matrix $T_{ij}$ via following relations:

$$P_i' = T_{ij} P_j, \tag{8}$$

where the transformation matrices for (001)-oriented and (111)-oriented thin films are given as follows:

$$
T_{ij}^{(001)} = \begin{pmatrix} 1 & 0 & 0 \\ 0 & 1 & 0 \\ 0 & 0 & 1 \end{pmatrix}, \;
T_{ij}^{(111)} = \begin{pmatrix} 0 & 1/\sqrt{2} & -1/\sqrt{2} \\ -2/\sqrt{6} & 1/\sqrt{6} & 1/\sqrt{6} \\ 1/\sqrt{3} & 1/\sqrt{3} & 1/\sqrt{3} \end{pmatrix}.
\tag{9}
$$

Similarly, elastic stiffness tensors, electrostrictive coefficient tensors, background dielectric constant tensors, and gradient energy coefficient tensors also need to be transformed from coordinate systems $\mathbf{x}$ to $\mathbf{x}'$ using transformation matrices via following relations:

$$c_{ijkl}' = T_{im} T_{jn} T_{ks} T_{lt} c_{mnst}, \tag{10}$$

$$Q_{ijkl}' = T_{im} T_{jn} T_{ks} T_{lt} Q_{mnst}, \tag{11}$$

$$\kappa_{ij}' = T_{im} T_{jn} \kappa_{mn}, \tag{12}$$

$$G_{ijkl}' = T_{im} T_{jn} T_{ks} T_{lt} G_{mnst}. \tag{13}$$

Here the temporal evolution of domain structures can be obtained by numerically solving the TDGL equation using the semi-implicit Fourier spectral method with previously reported materials parameters. We used discrete grid points of $256\Delta x \times 256\Delta y \times 40\Delta z$ with real grid space $\Delta x = \Delta y = \Delta z = 2$ nm to describe the film/substrate system, wherein the thickness of $PbZr_{0.2}Ti_{0.8}O_3$ is set as 48 nm with $h_{\text{film}} = 24\Delta z$ and the film/substrate mismatch strain $\bar{\varepsilon}_{ij}$ is set to be zero based on experimental observations. Using the superposition spectral method and short-circuit surface boundary conditions, we can solve the electrostatic equilibrium equation to obtain the energy evolution for polarization switching and domain structures in $PbZr_{0.2}Ti_{0.8}O_3$ films.

## Data availability
All data used in this manuscript are available from the authors on request.

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

## Acknowledgements

R.X. acknowledges support from the National Science Foundation under grant DMR-1708615. S.L. acknowledges the support from Carnegie Institution for Science. S.S. acknowledges support from the U.S. Department of Energy, Office of Science, Office of Basic Energy Sciences, under Award Number DE-SC-0012375 for the development of ferroelectric thin films. R.G. acknowledges support from the National Science Foundation under grant OISE-1545907. J.J.W. acknowledges partial support from the Army Research Office under grant number W911NF-17-1-0462. Z.J.H. and L.Q.C. acknowledge support from the U.S. Department of Energy, Office of Basic Energy Sciences, Division of Materials Sciences and Engineering under Award FG02-07ER46417. H.L. acknowledges support from the National Science Foundation under grant DMR-1608938. A.G. acknowledges support from the Gordon and Betty Moore Foundation's EPiQS Initiative, under grant GBMF5307. S.P. acknowledges support from the Army Research Office under grant W911NF-14-1-0104. E.B. acknowledges support from CAPES under Grant No. 9511/2014-08. Z.H.C. acknowledges support from the U.S. Department of Energy, Office of Science, Office of Basic Energy Sciences, Materials Sciences and Engineering Division under Contract No. DE-AC02-05-CH11231 (Materials Project program KC23MP) for the development of novel functional materials. A.M.R. acknowledges support from the Office of Naval Research, under Grant N00014-17-1-2574. L.W.M. acknowledges support from Intel Corp. as part of the FEINMAN program.

## Author contributions

R.X. and L.W.M. conceived of the study and designed the experiments. R.X. carried out the film synthesis, electrical measurements, and PFM characterization. S.L. performed the MD simulations. R.X., S.S., H.L., S.P. and E.B. conducted the capacitor fabrication and etching. J.W. and Z.H. performed the phase-field calculations. R.X., S.L., S.S., R.G., A.G., Z.H.C., L.Q.C., A.M.R. and L.W.M. analyzed the data and discussed the results. L.Q.C., A.M.R. and L.W.M. supervised the research efforts. All authors read and contributed during manuscript preparation.

## Additional information

**Competing interests:** The authors declare no competing interests.

**Journal Peer Review Information:** *Nature Communications* thanks Eric Bousquet and the other anonymous reviewers for their contribution to the peer review of this work. Peer reviewer reports are available.

