## [Peer Review File · Nature Communications]

Editorial Note: This manuscript has been previously reviewed at another journal that is not operating a transparent peer review scheme. This document only contains reviewer comments and rebuttal letters for versions considered at Nature Communications .

REVIEWERS' COMMENTS:

Reviewer #1 (Remarks to the Author):

The authors have extendedly addressed my comments. As I wrote in my previous review I have no doubts on the experimental side of the manuscript. My main concern is the sufficient elements of novelty. In the literature on memristive devices variety of materials and switching mechanisms were offered within the recent 10 years (including ferroelectric memories), where the switching mechanisms may vary even in the same system or device, depending on internal or external factors. Nowadays most reasonable contribution is not explaining one particular mechanism in one particular system, but to find a reliable system in respect performance, that is worth of investigating the mechanisms. In my general opinion this manuscript should have been contributing more reliably on the performance to merit publication in Nature journals. On the other hand the manuscript shows a strong fundamental aspect, which I find in this case sufficiently important to recommend publication in Nature Communications.

Reviewer #2 (Remarks to the Author):

The authors addressed all the earlier comments in a very satisfactory way. The paper is well written, presents interesting switching phenomena and can be published as it is.

Reviewer #3 (Remarks to the Author):

One of the main comment (from myself and Reviewer 2) was regarding the novelty of the methods and results reported in the present paper. The first main response is that they show the possibility to control, assess and stabilise multiple polarisation states related to the the switching "procedure" (i.e. field amplitudes) used and so related to different competing switching pathways. I understand their clarification and agree that on this aspect the results reported are indeed new and go beyond the ones reported in their previous paper. Being able to stabilise multiple polarisation states in ferroelectric materials is clearly of high interest and appealing.

The second main response is that their new results allows to understand better the physics of multi-steps switching mechanisms of ferroelectrics since they were able to propose explanations on the origin of the kinetic vs stochastic routes for switching the polarisation. This is done mostly through the simulation part of the paper where the decomposition of the energy between its different contributions (mostly depolarising field and elastic contributions), which allows to give more conclusion about their effect on high/low field switching mechanism of the domain states. I thus think that the authors clarified and updated the text accordingly to clarify how novel and important their new works are through a combined experimental observation/microscopic explanations through simulations, which, indeed, can be published in Nature Materials (and as I said, the quality of the works and manuscript is clearly high).

My last comment was about the proof that the 111 domain orientation is indeed a key point to observe the reported behaviour of controlling and stabilise the different multiple polarisation states in ferroelectrics. I understand that the proof of it is certainly beyond the scope of the present paper (I was mentioning BaTiO₃ in case the authors have a model for it and can check quickly to improve the impact of the paper) but I would explicitly say that it will be an important point to prove in the future (this is just a proposition not "mandatory").

This last point was making me hesitating to accept this paper for Nature Materials or moving it to Nature Communication, however, after the discussion made in their response letter for the aforementioned main comment, I think the paper indeed deserves to be published in Nature Materials.

Reviewer #4 (Remarks to the Author):

In this paper, the authors introduce the realisation of multiple ferroelectric states with intermediate polarisation values in (111)-oriented $\text{PbZr}_{0.2}\text{Ti}_{0.8}\text{O}_3$ films. It is based on the control of two different switching pathways that are dependent on the height of the applied electric field, e.g. for higher fields, bipolar switching is favoured, for lower fields multi-step switching occurs. The authors use different methods, e.g. molecular dynamics simulations, PFM measurements and phase-field modelling to create a full model describing the switching process under different electric fields.

The paper is clearly structured and yields interesting information regarding the visualization of the different polarisation states. However, I have concerns regarding the publication in Nature communications. A few comments are listed below:

Most importantly, there is already a previous publication of the authors (Nature Mater. 14, 79-86 (2015), where the same material system is studied using the same methods (PFM, MD simulations). In this previous paper, the principle of multi-step 90° switching is already visualized with PFM measurements. It is not clear at all what major step forward was made by the authors which could justify another publication in Nature Mater.

Regarding the context of applications:

It is already known from FeRAM research that down-scaling of PZT is not possible without the loss of ferroelectric properties and the formation of monodomains.

The device thickness of 100 nm is not competitive compared with other emerging memory technologies and not suited for future applications.

In the paper, the retention of the device seems to be quiet ok due to the oxide electrodes, but no statement about the imprint effect is made which could be a main disadvantage and lead to a shift of the switching voltage.

In the abstract, the possible application for neuromorphic devices is mentioned and shortly taken up in the last sentences of the summary. However, during the main part of the paper no reference to neuromorphic switching is made. Additional experiments or explanations are missing.

In summary, if at all, I would recommend the paper for another journal. Perhaps SREP is a good choice since they do not insist on full novelty.

Overall Response to Reviewers and Editors

We thank the Reviewers for their time in assessing our manuscript and for the valuable comments that they have provided. Here, we provide detailed, point-by-point responses that, we believe, carefully address each individual comment. Based on this, we have amended and improved our manuscript keeping these comments and suggestions in mind, and also include additional changes as recommended by the editor.

Response to Reviewer #1

Comment: “The authors have extendedly addressed my comments. As I wrote in my previous review I have no doubts on the experimental side of the manuscript. My main concern is the sufficient elements of novelty. In the literature on memristive devices variety of materials and switching mechanisms were offered within the recent 10 years (including ferroelectric memories), where the switching mechanisms may vary even in the same system or device, depending on internal or external factors. Nowadays most reasonable contribution is not explaining one particular mechanism in one particular system, but to find a reliable system in respect performance, that is worth of investigating the mechanisms. In my general opinion this manuscript should have been contributing more reliably on the performance to merit publication in Nature journals. On the other hand the manuscript shows a strong fundamental aspect, which I find in this case sufficiently important to recommend publication in Nature Communications.”

Response: We thank the Reviewer for theses valuable comments and the recommendation of publication.

Response to Reviewer #2

Comment: “The authors addressed all the earlier comments in a very satisfactory way. The paper is well written, presents interesting switching phenomena and can be published as it is.”

Response: We appreciate the Reviewer’s assessment about the quality of our results and writing and thank the Reviewer for the recommendation of publication.

Response to Reviewer #3

Comment: “One of the main comment (from myself and Reviewer 2) was regarding the novelty of the methods and results reported in the present paper. The first main response is that they show the possibility to control, assess and stabilise multiple polarisation states related to the the switching “procedure” (i.e. field amplitudes) used and so related to different competing switching

pathways. I understand their clarification and agree that on this aspect the results reported are indeed new and go beyond the ones reported in their previous paper. Being able to stabilise multiple polarisation states in ferroelectric materials is clearly of high interest and appealing.

The second main response is that their new results allows to understand better the physics of multi-steps switching mechanisms of ferroelectrics since they were able to propose explanations on the origin of the kinetic vs stochastic routes for switching the polarisation. This is done mostly through the simulation part of the paper where the decomposition of the energy between its different contributions (mostly depolarising field and elastic contributions), which allows to give more conclusion about their effect on high/low field switching mechanism of the domain states. I thus think that the authors clarified and updated the text accordingly to clarify how novel and important their new works are through a combined experimental observation/microscopic explanations through simulations, which, indeed, can be published in Nature Materials (and as I said, the quality of the works and manuscript is clearly high).

My last comment was about the proof that the 111 domain orientation is indeed a key point to observe the reported behaviour of controlling and stabilise the different multiple polarisation states in ferroelectrics. I understand that the proof of it is certainly beyond the scope of the present paper (I was mentioning BaTiO₃ in case the authors have a model for it and can check quickly to improve the impact of the paper) but I would explicitly say that it will be an important point to prove in the future (this is just a proposition not “mandatory”).

This last point was making me hesitating to accept this paper for Nature Materials or moving it to Nature Communication, however, after the discussion made in their response letter for the aforementioned main comment, I think the paper indeed deserves to be published in Nature Materials.”

Response: We appreciate the Reviewer’s assessment about the quality of our results and writing and thank the Reviewer for the recommendation of publication. We agree with the Reviewer that investigating similar switching behaviors in other material systems such as (111)-oriented BaTiO₃ films will be an important future direction. This will facilitate the understanding whether (111)-orientation triggers this type of switching behavior in other ferroelectric systems.

Response to Reviewer #4:

Comment: “In this paper, the authors introduce the realisation of multiple ferroelectric states with intermediate polarisation values in (111)-oriented PbZr_{0.2}Ti_{0.8}O₃ films. It is based on the control of two different switching pathways that are dependent on the height of the applied electric field, e.g. for higher fields, bipolar switching is favoured, for lower fields multi-step switching occurs. The authors use different methods, e.g. molecular dynamics simulations, PFM measurements and phase-field modelling to create a full model describing the switching process under different electric fields. The paper is clearly structured and yields interesting information regarding the visualization of the different polarisation states. However, I have concerns regarding the publication in Nature communications.”

Response: We thank the Reviewer for the detailed reading and these valuable comments and feedback. In the following we attempt to address any potential concerns.

Comment: “A few comments are listed below: Most importantly, there is already a previous publication of the authors (Nature Mater. 14, 79-86 (2015), where the same material system is studied using the same methods (PFM, MD simulations). In this previous paper, the principle of multi-step 90° switching is already visualized with PFM measurements. It is not clear at all what major step forward was made by the authors which could justify another publication in Nature Mater.”

Response: To begin, we are no longer talking about publication in *Nature Materials*, so perhaps this is solved query. More generally though, we do respectfully disagree with the Reviewer’s assessment that the current work has already been visualized in *Nature Mater.* **14**, 79-86 (2015) and believe that this arises from potential confusion about various terms and the important differences in their meanings. Although the two papers studied similar material systems and utilized similar methods, both the main observations/findings and the underlying physics in the two papers are completely different. In the following we summarize and differentiate the two papers:

Summary of Nature Mater. 14, 79-86 (2015) – The focus of this paper was on a nanoscale study of local switching events in single ferroelectric domains in (001)- and (111)-oriented $\text{PbZr}_{0.2}\text{Ti}_{0.8}\text{O}_3$ films. The main observation and conclusion of this work was that traditional *single-step 180° ferroelectric switching* is favored in (001)-oriented $\text{PbZr}_{0.2}\text{Ti}_{0.8}\text{O}_3$ thin films, while *successive 90° ferroelastic switching* is favored in (111)-oriented $\text{PbZr}_{0.2}\text{Ti}_{0.8}\text{O}_3$ thin films (Fig. R2).

Summary of the current work – Although we again compare (001)- and (111)-oriented $\text{PbZr}_{0.2}\text{Ti}_{0.8}\text{O}_3$ thin films, the current manuscript contains a number of observations and conclusions that are markedly different from the prior work and represent the foundation of the novelty of this approach. These novelties and differences are summarized here:

- 1) “*Multivalued character*” – The focus of our current manuscript is on how to kinetically control different mesoscale domain structures so as to access *multi-state polarizations*. To directly address the Reviewer’s comment, the “multivalued character” in these two papers is *fundamentally different*. The “multivalued character” in the 2015 *Nature Mater.* paper is a multi-step switching process; the “multivalued character” in the current manuscript concerns multi-state polarizations, resulting from the kinetic control of two competing switching pathways which give rise to different domain configurations during partial switching and, in turn, realization of stable, partially-switched polarization states.
- 2) *Observation of multiple, competing switching pathways* – To date, essentially all work on ferroelectrics has assumed or observed a one-to-one correspondence between the switching pathway and the material system – including our prior work in the 2015 *Nature Mater.* paper

Fig. R2. The schematic of local switching events in (001)- and (111)-oriented $\text{PbZr}_{0.2}\text{Ti}_{0.8}\text{O}_3$ thin films.

noted here (direct single-step 180° , ferroelectric switching in (001)-oriented films and multi-step 90° , ferroelastic switching in (111)-oriented films). One of the key novel observations of the current manuscript, is that we observe *two different, and competing, switching pathways* in (111)-oriented $\text{PbZr}_{0.2}\text{Ti}_{0.8}\text{O}_3$ thin films. These different pathways include the multi-step 90° , ferroelastic switching under low fields (Fig. 4f; which was observed in the 2015 *Nature Mater.* paper), and the *bi-polar-like, direct 90° ferroelastic switching* under application of large fields (which has not been observed before; Fig. 4g). Such a *kinetic competition* between polarization switching processes has not been reported and, in turn, provides for unprecedented function in these materials.

- 3) *Improved understanding of the multi-step switching mechanism at low-fields* – With the combination of PFM measurements and MD simulations, we now have an improved understanding of the multi-step 90° switching at low fields. We identified two types of 90° *local* switching events at low fields: 1) switching wherein there is a change in the out-of-plane component of polarization (e.g., $P_3^- \rightarrow P_1^+$, Fig. 4; henceforth referred to as *sign-changing switching*) and 2) switching wherein there is no change in the out-of-plane component of polarization (e.g., $P_1^- \rightarrow P_3^-$, Fig. 4; henceforth referred to as *sign-conserving switching*). Energetically, sign-changing switching is favored *thermodynamically* by an electrostatic energy gain $-EP$ but is associated with an elastic energy barrier ΔE_e^1 while sign-conserving switching is associated only with an elastic energy barrier ΔE_e^2 . Moreover, we find that $\Delta E_e^1 \gg \Delta E_e^2$ because the transient state of sign-changing switching is fully within the (111) and thus subject to stronger clamping effects whereas the transient state of sign-conserving switching has a smaller in-plane component of polarization (Fig. 4i). Thus, at low fields (where the elastic energy dominates), as *sign-changing* switching occurs first in one domain, the neighboring domain compensates the strain via a *sign-conserving* switching event, which is much more rapid *kinetically* due to the lower elastic-energy barrier. This helps rationalize the formation of the 50% poled *Type-II* twinning structure with half of the domains up-poled (resulting from the *sign-changing* switching) and half of the domains down-poled (resulting from the *sign-conserving* switching). The net-zero out-of-plane polarization of the *Type-II* twinning structure also helps stabilize this structure, due to the absence of a depolarization field. This insight was not developed or understood before.
- 4) *Novel bipolar-like, direct 90° ferroelastic switching mechanism at high fields* – This high-field switching mechanism is a completely new discovery. Although the pulsed switching measurement reveals a polarization-time profile similar to that of traditional 180° switching in (001)-oriented thin films, the polarization reversal, as suggested by MD simulations, is realized by coordinated sign-changing switching events (e.g., simultaneous $P_1^- \rightarrow P_3^+$ and $P_3^- \rightarrow P_1^+$; Fig. 4g). This switching mechanism does not involve significant changes in the overall twinning structure orientation and is a newly identified domain reversal switching mechanism.
- 5) *Unified understanding of the switching mechanisms* – We now have a unified understanding of the field-strength-dependent switching mechanisms. To summarize, at low fields, when the elastic energy dominates, the system proceeds through sign-conserving switching events much more rapid because of the lower elastic energy barrier. As a result, we observe the multi-step 90° , ferroelastic switching pathway with the change in twinning structures. At high fields, when the electrostatic energy dominates, the system prefers proceed through more sign-changing switching events that have larger electrostatic energy gain. As a result, we observe a bipolar-like, direct 90° ferroelastic switching accomplished through coordinated sign-changing switching without substantial changes in the twinning structures. When the electric field is

intermediate, such that the electrostatic energy EP is comparable to $\Delta E_e^1 - \Delta E_e^2$ (Bell–Evans–Polanyi principle), both switching pathways are comparably fast, producing a mixture of *Type-I* and *-II* twinning structures and different intermediate polarization states. This is important new observation and physics and is added in the revised manuscript.

- 6) *Kinetic vs. stochastic route to multi-states* – Prior work, as summarized in the manuscript, has attempted to make use of the bipolar stability of ferroelectrics to produce multi-state function. As noted, however, this work has relied on a stochastic process wherein partially-switched polarization states must be stabilized by the presence of defects or other pinning potentials. This is the case for the data on (001)-oriented films provided in our current work, and as a result, the repeatability and stability of these intermediate states are found to be lacking. The (111)-oriented $\text{PbZr}_{0.2}\text{Ti}_{0.8}\text{O}_3$ thin film provides a way to move beyond defect-mediated pinning to produce stable, deterministic multi-state polarization. The novelty here lies in the fact that the competing domain switching processes result in mixtures of *Type-I* and *-II* twinning structures. This does not require defects or other pinning potentials, but represents an intrinsic feature of the material wherein intermediate stable states are produced.

All told, the work in this current manuscript dramatically moves beyond the observations and understanding provided in the 2015 *Nature Mater.* paper.

Comment: “Regarding the context of applications: It is already known from FeRAM research that down-scaling of PZT is not possible without the loss of ferroelectric properties and the formation of monodomains. The device thickness of 100 nm is not competitive compared with other emerging memory technologies and not suited for future applications. In the paper, the retention of the device seems to be quiet ok due to the oxide electrodes, but no statement about the imprint effect is made which could be a main disadvantage and lead to a shift of the switching voltage.”

Response: We do not disagree with the Reviewer, the reported device performance (*i.e.*, scalability, retention, imprint) does not exhibit apparent advantages over other existing FeRAM devices. This said, we would like to emphasize that this work is not focused on the realization of a new “device” to replace such systems, but is primarily focused on the identification of novel materials phenomena, namely the realization of tunable, multi-state polarization via control of competing switching processes. The results, in turn, represent an interesting set of phenomena, not the demonstration of an optimized device platform. At the current stage, we are not aiming or contending to present a paradigm shift in device applications, but focus more on the mechanism of multi-state switching phenomena.

In addition, although scalability is not the focus of our current manuscript (which is focused more on the novel switching phenomena and less on engineering an optimized device), we believe that there are ways to make the device scalable both at the material and device levels. For instance, at the material level, the scalability of multi-state switching is limited by the length scale of the mesoscale domain structures, which in other words is limited by the size of the nano-domains. Following Kittel’s law, the domain size scales with the square root of the film thickness, thus reducing the film thickness will decrease the nano-domain size to make the device more scalable and compatible with smaller device structures. At the device level, the switching process studied herein can readily be integrated into and optimized in a number of device architectures. Foremost of which are either a ferroelectric-gated field-effect transistor (FeFET) or a ferroelectric tunnel junction (FTJ). For instance, FeFETs are highly scalable and provide for non-volatile and non-

destructive reading, but have often been limited by retention issues. Thus, the (111)-oriented $\text{PbZr}_{0.2}\text{Ti}_{0.8}\text{O}_3$ films studied in the current work are an important step forward for such devices since we observe the ability to create multiple polarization states in a deterministic and repeatable manner, in a system that also exhibits improved retention and endurance properties. On the other hand, FTJs also demonstrates excellent scalability. Although, not within the scope of the current work, studying ultra-thin (111)-oriented $\text{PbZr}_{0.2}\text{Ti}_{0.8}\text{O}_3$ based FTJs is a promising future direction, as all of the other characteristics of this material system make it an excellent candidate to provide the function desired for non-volatile memory applications.

Comment: “In the abstract, the possible application for neuromorphic devices is mentioned and shortly token up in the last sentences of the summary. However, during the main part of the paper no reference to neuromorphic switching is made. Additional experiments or explanations are missing.”

Response: We apologize for this confusion. We have removed reference to neuromorphic switching in our manuscript.

Comment: “In summary, if at all, I would recommend the paper for another journal. Perhaps SREP is a good choice since they do not insist on full novelty.”

Response: We respect the Reviewer’s opinion, but we believe our response to Reviewer’s comment regarding the difference between our current work and the work in 2015 *Nature Mater.* paper has clarified the unique features of our current work, which supports the argument of novelty of our current work.